# Comparative Preclinical Study of Lidocaine and Mepivacaine in Resilient Hyaluronic Acid Fillers

**DOI:** 10.3390/pharmaceutics14081553

**Published:** 2022-07-26

**Authors:** Romain Brusini, Julien Iehl, Elodie Clerc, Mélanie Gallet, François Bourdon, Jimmy Faivre

**Affiliations:** Research and Development Department, Teoxane SA, Rue de Lyon 105, 1203 Geneva, Switzerland; r.brusini@teoxane.com (R.B.); j.iehl@teoxane.com (J.I.); e.clerc@teoxane.com (E.C.); m.gallet@teoxane.com (M.G.); f.bourdon@teoxane.com (F.B.)

**Keywords:** hyaluronic acid, anesthetics, soft-tissue fillers, lidocaine, mepivacaine, release, pharmacokinetics

## Abstract

Background: Hyaluronic acid-based filler injections are now well-established aesthetic procedures for the correction of skin tissue defects and volume loss. Filler injections are becoming increasingly popular, with a growing number of injections performed each year. Although classified as a minimally invasive procedure, the introduction of a needle or a canula may remain painful for the patient. A major improvement was achieved with the incorporation of local anesthetics into the formulation for pain relief. Methods: In this study, two well-known anesthetics, lidocaine and mepivacaine, were systematically compared to assess their influence on filler mechanical and biological features. The impact of each anesthetic was monitored in terms of gel rheological properties, stability, durability, and degradation. The release profiles of each anesthetic were also recorded. Finally, the pharmacokinetics of each anesthetic in rats were assessed. Results: For all the rheological and biological experiments performed, lidocaine and mepivacaine influences were comparable. The addition of either anesthetic into a soft-tissue filler showed no significant modifications of the stability, durability, and degradability of the gel, with similar release profiles and pharmacokinetics at an equivalent concentration. Conclusions: Substituting lidocaine with mepivacaine does not impact the properties of the gels, and thus both can be equally incorporated as anesthetics in soft-tissue fillers.

## 1. Introduction

Nowadays, popular rejuvenation treatments of skin depressions and wrinkles in aesthetic practice involves injections of soft-tissue fillers, including mostly crosslinked hyaluronic acid (HA)-based gels, within different skin tissue layers and at different locations via the aid of needles or cannulas. Even though this nonsurgical procedure is characterized as minimally invasive, the injection may nevertheless remain painful for the patient, particularly in certain indications such as lips remodeling, due to facial tissue disruption generating nerve responses. Pain management is therefore a key point for clinicians to satisfy their patients during the procedure [1,2]. A substantial evolution of HA fillers came in the 2000s with the introduction of anesthetics in the gel formulations to minimize discomfort to the patient [2].

Among the library of local anesthetics, amino ester local anesthetics were first synthesized between 1891 and 1930, such as tropocaine, eucaine, holocaine, orthoform, benzocaine, and tetracaine [3], but these local anesthetics were often associated with poor stability due to the hydrolysis of the ester linkage [4]. Later, the amino–amide family of anesthetics, including lidocaine, articaine, mepivacaine, bupivacaine, and ropivacaine, emerged with enhanced chemical stability [4] and are still commonly used for pain management during minor surgery or more invasive procedures.

Amino–amide anesthetics are classically composed of a lipophilic aromatic ring and a hydrophilic tertiary amine linked together with an amide bond [5,6]. Lidocaine, the first amino–amide-type local anesthetic, was first synthesized under the name ‘xylocaine’ by Swedish chemist Nils Löfgren in 1943. Later, in 1957, af Ekenstam et al. synthesized mepivacaine and bupivacaine, both very similar in structure and associated with longer anesthetic durations than lidocaine [7]. With the exception of articaine, each amino–amide anesthetic preserved the 2,6-xylidine aromatic ring and varied in the tertiary amine with a diethylamino moiety for lidocaine and an alkyl-piperidine ring with, respectively, a methyl for mepivacaine, a propyl for ropivacaine, and a butyl moiety for bupivacaine. Novel local anesthetics, such as liposomal preparations, animal toxins, vanilloids, or polymers, are currently in development [8].

The mechanism of action of these local anesthetics lies in the reversible and concentration-dependent inhibition of sensory neuronconduction via the inactivation of voltage-gated sodium channels in the neuronal membrane [9,10]. By this process, nociceptive fibers’ depolarization is inhibited, which in consequence prevents the afferent transmission of pain impulses [8]. Topical anesthetic mixtures or nerve blocks may be used prior to soft-tissue filler injection procedures to further improve pain management, but to date, there are no clear guidelines [11]. Since 2005, lidocaine has been successfully introduced at 0.3% in weight into HA fillers and is now commercially used worldwide. A recent meta-analysis compiled the pain, effectiveness, and safety outcomes for the treatment of nasolabial folds (NLF) with HA fillers in presence or absence of 0.3% *w/w* lidocaine extracted from 12 randomized clinical trials. The pain, assessed using the visual analogue scale (VAS), was lower in the lidocaine group whereas the clinical effectiveness of the NLF treatment, assessed using the wrinkle severity rating scale (WSRS), or the safety of the procedure reported in frequency of mild and transient adverse events such as injection site swelling, erythema, bruising, itching, or induration, did not exhibit any statistical differences [12]. Similarly, other amino–amide anesthetics, such as articaine, mepivacaine, and bupivacaine, have already been tested for dental anesthesia [13,14,15] and could be envisaged into HA fillers as local anesthetics.

In this study, mepivacaine was investigated as a potential new local anesthetic agent to be used in HA fillers to replace lidocaine. The choice of mepivacaine is notably encouraged by its lower vasodilatory activity compared to lidocaine, with mepivacaine tending to either preserve or decrease peripheral blood flow, keeping a lower systemic concentration of mepivacaine over time, which would turn out to be an additional safety aspect [16,17,18]. The influence of this anesthetic agent was then compared to the gold standard lidocaine in terms of filler mechanical properties, stability, degradability, release profiles, and pharmacokinetics. Such data are especially useful for a better evaluation of the safety and the performance of mepivacaine-loaded fillers compared to their lidocaine counterparts and, as such, are relevant for the registration of the device.

## 2. Materials and Methods

### 2.1. Hyaluronic Acid-Based Fillers

HA-based fillers PNT-1, intended to correct superficial fine lines, and PNT-4, used subcutaneously or in deep fat compartments as a volumizer, were chosen among the Teosyal RHA**^®^** collection manufactured by Teoxane SA, Switzerland. Teosyal RHA**^®^** fillers are manufactured using the Preserved Network Technology (PNT) aimed to ensure mild HA crosslinking conditions to lower their degradation compared to conventional crosslinking processes and enable low amounts of crosslinker to be used [19,20]. The characteristics and the intended use of PNT-1 and PNT-4 are presented in Table 1. Lidocaine HCl or mepivacaine HCl were added to each formulation at the concentration of 0.3% wt/wt for comparison purpose. The gels were manufactured following identical processes and steam sterilized prior to further use.

The rheological characteristics of the fillers were assessed using a DHR-2 rheometer (TA Instruments, New Castle, DE, USA). The elastic modulus and the phase angle were measured with a cone–plate geometry (1°, stainless steel, 40 mm diameter, 24 µm gap) at 1 Hz and a stress of 5 Pa at 25 °C using 0.5 g of gel. The Stretch score was determined using a creep test and measured with a rough parallel plate geometry (stainless steel, 25 mm, 500 μm gap) under a constant stress of 5 Pa at 25 °C over 15 min. The injection force was measured using a force tester MultiTest-dV (Mecmesin, Slinfold, UK,) using 30 G ½’ and 27 G ½’ needles for PNT-1 and PNT-4, respectively, at a constant speed of 12.5 mm/s. Every measurement was performed three times in the same batch.

### 2.2. Shelf-Life Stability Studies of HA-Based Fillers in Presence of Lidocaine or Mepivacaine

The shelf-life stability of sterilized PNT-1 and PNT-4 formulated either with lidocaine or mepivacaine was assessed in a controlled-atmosphere chamber at 25 °C and 60% relative humidity. As a product specification, the phase angle, δ, was monitored over 36 months. At each timepoint, 3 syringes from different batches were collected and rheologically assessed using a DHR-2 rheometer (TA Instruments, New Castle, USA) equipped with a cone–plate geometry (1°, stainless steel, 40 mm diameter, 24 µm gap). The phase angle δ was measured at 1 Hz and a stress of 5 Pa at 25 °C.

### 2.3. Release Profiles of Lidocaine and Mepivacaine from HA-Based Fillers

The release test was performed using a method developed in accordance with USP 1724—semisolid drug products—performance test. Immersion cells (model B) placed in 150 mL vessels and assembled with USP apparatus 2 (Dissolutest Hanson Vision Elite 8, Teledyne Instruments, Chatsworth, CA, USA) were filled with 600 mg of gel (1.8 mg of anesthetic in each cell). Gel surface was covered with a hydrophilic 0.45 µm polyethersulfone membrane (Supor^®^ 450, Pall, Basel, Switzerland). Vessels were filled at 150 mL with phosphate buffer at pH 7.3 to ensure sink conditions and the set up was thermostatically controlled at 37 °C and stirred at 100 rpm. An amount of 1.5 mL of release medium were retrieved after 15, 60, 120,180, 240, 300, and 360 min and released lidocaine or mepivacaine were quantified by HPLC-UV at 230 nm. The HPLC setup (Hitachi, Tokyo, Japan) was composed of an XBridge Shield RP18 column (Waters, Milford, CO, USA) and a phosphate buffer/acetonitrile 50/50 mobile phase set a flow rate of 1.0 mL/min. Each test was performed in triplicate.

### 2.4. Degradation of Gels in Presence of High and Low Doses of Hyaluronidase

The gel in vitro kinetics of degradation were monitored over time using a DHR-2 rheometer (TA Instruments, New Castle, DE, USA) and in the presence of hyaluronidase. The loss of the viscoelastic properties of the gels, for instance the elastic modulus, G′, were followed until complete degradation of the gels. By varying the number of units of hyaluronidase, it was possible to either investigate the gel capacity to readily degrade in case of an adverse event (fast degradation test with high doses of enzyme) or the gel persistence (persistence test with low doses of enzyme). For the fast degradation test, 1 g of PNT-1 or 0.5 g of PNT-4 was homogenized with 50 µL of hyaluronidase (Hyaluronidase 1500 I.E., Wockhardt UK Ltd., Wrexham, UK, 70 U/g for PNT-1 and 140 U/g for PNT-4) through 20 successive extrusions in syringes interconnected with a luer-lock. The gel was then equilibrated at 37 °C between a cone–plate geometry (anodized aluminum, 1°, 40 mm, 24 µm gap) for 1 min before the measurement of G′ over time at a stress of 5 Pa and a frequency of 1 Hz. Every 5 min, the hyaluronidase solution was refreshed with a new 50 µL aliquot added on top of the gel until complete degradation. The same protocol was implemented for the persistence test except that a lower activity of enzyme was introduced every 5 min (Hylase “Dessau” 150 I.E., Riemser Pharma GmbH, Greifswald, Germany, 7 U/g of gel) to observe any difference in the gel’s mechanical performances over time. The gels were considered as fully degraded once their G′ dropped under 30 Pa. Below this limit, the gels were liquefied and not properly assessed in the conditions of the measurement. Each test was performed in triplicate.

### 2.5. In Vivo Investigations

#### 2.5.1. Animals

PNT-4 gels with either lidocaine hydrochloride or mepivacaine hydrochloride were injected once intradermally in 9–11-week-old male and healthy Sprague–Dawley rats (*n* = 24, 355–430 g, Charles River, Sulzfeld, Germany) at a concentration of 1 g/kg and at 6 different locations on their back. The experiment was performed in an AAALAC-accredited laboratory and in accordance with German animal protection law, subjected to Ethical Review Process, accepted by local authority, and authorized by the Bavarian animal welfare administration. The animals were housed in groups of 3 in a humidity, temperature-controlled, and air-conditioned individually ventilated cages under artificial light, with a 12/12 h light/dark cycle. The animals were acclimated at least for 5 days under laboratory conditions and had free access to standard food and water. During the course of the experiment, animals were daily checked for any clinical sign of toxicity.

#### 2.5.2. Pharmacokinetics of Lidocaine and Mepivacaine in Rats after Intradermal Filler Injection

Two groups (*n* = 12 for each group) of rats were injected with the 2 PNT-4 formulations containing either lidocaine or mepivacaine. Each group was divided into 4 subgroups (*n* = 3 for each subgroup) for the different blood sampling timepoints (3 timepoints for each subgroup). Each subgroup was used for 3 blood sampling timepoints from the sublingual vein under slight anesthesia (isoflurane) right before injection and at 5, 30 min, 1, 2, 4, 6, 12, 24, and 48 h. An amount of 500 µL of blood were sampled in K2-EDTA-coated tubes, centrifuged for 10 min at 4 °C and 1000× *g*. After centrifugation, 140 µL of plasma was retrieved and plasma concentrations of lidocaine and mepivacaine were analyzed using a validated LC (Acquity UPLC I-Class, Waters, Milford, CO, USA) MS/MS (Xevo TQ-S, Waters, Milford, CO, USA) method. Pharmacokinetic parameters, including time to reach max concentration (Tmax) and elimination half-life (t_1/2_), were calculated using a noncompartmental analysis for extravascular administration and obtained using Phoenix WinNonLin^®^ software (version 8.0, Certara, Princeton, NJ, USA).

## 3. Results

### 3.1. HA Fillers

The mechanical properties of the studied fillers were assessed as a first readout after final packaging and sterilization within syringes. The elastic modulus, G′, characterizing the elastic behavior of the gel in nearly static conditions and the phase angle, δ, probing the ratio of the viscous and the elastic behaviors of the gels (viscous/elastic ratio) are reported in Table 2, as well as the Strength and the Stretch scores [19]. The Strength is a two-dimension parameter which probes the elastic modulus G′ and the range of deformations or stresses the gel can withstand without losing its structure and fully recover, namely, the Linear Viscoelastic Region (LVER). Meanwhile, the Stretch scores the rate of the deformation of a gel submitted to a stress, i.e., the higher the Stretch is, the more the gel may accommodate more naturally and rapidly to facial expressiveness. The Strength score as well as the G′ were higher for PNT-4 than PNT-1, meaning that PNT-4 presented a higher elastic behavior, suitable for deeper indications to lift tissues. PNT-1, on the other hand, presented a higher Stretch score compared to PNT-4, which confirmed that PNT-1 is well adapted to the superficial dynamic areas of the skin. Upon comparison of lidocaine and mepivacaine, no statistical variations were observed between both formulations for either PNT-1 or PNT-4, and, moreover, they remained well within the products’ specifications, meaning that the addition of 0.3% lidocaine or mepivacaine within the commercial fillers PNT-1 and PNT-4 did not impact the rheological properties of each product. Thus, the addition of either anesthetic does not impair their balanced mechanical characteristics (softness and stretchability for PNT-1, stiffness and cohesivity with a large LVER for PNT-4), making them adapted for dynamic areas of the face. In addition, both gels exhibited low extrusion forces independently of the use of lidocaine or mepivacaine, which should make them easy to inject by the healthcare practitioners.

### 3.2. Shelf-Life Stability Studies of HA-Based Fillers in Presence of Lidocaine or Mepivacaine

The shelf-life stability of the products in presence of lidocaine or mepivacaine showed identical trends over a storage period of 3 years (Figure 1). The key rheological parameter phase angle, δ, did not show any significant differences between both the lidocaine and mepivacaine groups. Moreover, the phase angles remained well within the product specifications of each filler over three years, which guarantee to the practitioner and the patient the same clinical outcomes within the use-by date.

### 3.3. Release Study of Lidocaine and Mepivacaine from HA-Based Fillers

The in vitro release profiles of lidocaine and mepivacaine from PNT-1 and PNT-4 were monitored over time in phosphate buffer (Figure 2). All curves presented similar shapes, with all mepivacaine curves trending slightly steeper than lidocaine’s, suggesting a potential clinical benefit of mepivacaine-containing gels of a faster access to the anesthetic effects. Nevertheless, each datapoint was proven to be not significantly different from each other using a Fisher test F2, meaning that the lidocaine and mepivacaine release profiles were considered equivalent. All formulations almost reached complete release after 6 h in the conditions of the test.

### 3.4. Degradation of Gels in Presence of High and Low Doses of Hyaluronidase

Two experiments were conducted in parallel: a fast degradation test in the presence of high doses of enzyme and a gel persistence test in the presence of low doses of enzyme. The purpose of the fast degradation study was to observe any impact of lidocaine or mepivacaine on the propensity of the gels to readily degrade in the presence of high doses of hyaluronidase, mimicking a potential need to dissolve the gel in the case of serious adverse events such as vascular occlusion, in case of overcorrection, or in case of undesired aesthetic results. The fast degradation data are presented in Figure 3A. Both PNT-1 and PNT-4 exhibited fast degradation within 5 min in the presence of a saturated dose of hyaluronidase, and no differences were observed between the lidocaine and mepivacaine formulations. It is noteworthy that the G′ values of each gel started well below their initial G′ values due to the fast degradation occurring during the incorporation of the enzyme and the temperature equilibration time of the rheometer. The persistence test was subsequently carried out in the presence of low, repeat doses of hyaluronidase (Figure 3B). This test might mimic the long-term degradation of gels in presence of endogenous hyaluronidase, but in an in vitro accelerated manner, to distinguish potential differences. As expected, PNT-4 presented a slower degradation than PNT-1 consistently with its higher HA concentration, degree of modification, and, as a consequence, higher clinical duration [19,21,22,23]. Once again, the use of lidocaine or mepivacaine did not influence the persistence of the investigated gels. Under the same conditions of degradation, even though they started from separated production batches, PNT-4 with lidocaine fully degraded after 37.7 ± 3.2 min, whereas PNT-4 with mepivacaine degraded after 36.3 ± 1.4 min. Similarly, PNT-1 with lidocaine lasted 11.2 ± 1.4 min, whereas PNT-1 with mepivacaine lasted 10.7 ± 0.7 min.

### 3.5. Phamarcokinetics of Lidocaine and Mepivacaine after Intradermal Filler Injection in Rats

Prior to the in vivo assessment of the gels with the different anesthetics, PNT-1 and PNT-4 gels with lidocaine or mepivacaine were tested on- L-929 Mouse Fibroblast cells to determine their cytotoxicity via an MTS assay according to the ISO 10993-5 guidelines (data not shown). None of the investigated gel formulations presented toxicity towards fibroblasts, and importantly, no statistical differences were observed between both lidocaine and mepivacaine groups.

The pharmacokinetic behavior of lidocaine and mepivacaine were assessed after intradermal injection of the gels in rats. Neither mortality nor any clinical findings were noticed throughout the period of the treatment in both groups. The mean body weights of the animals remained stable, highlighting the tolerance of the treatments. The plasma concentrations of both anesthetics showed a rapid increase and decrease (Figure 4). The calculated pharmacokinetic parameters related to the lidocaine and mepivacaine treatments showed a slightly shorter absorption (Tmax) and a greater Cmax (Table 3) for lidocaine. The maximal concentration was obtained at 30 min for lidocaine and 1 h for mepivacaine. Oppositely, the AUC was slightly higher for mepivacaine than for lidocaine, demonstrating a greater exposure to mepivacaine. After 24 h, the plasmatic concentrations of both groups were below the lower levels of quantification set at 50 and 20 pg/mL for lidocaine and mepivacaine, respectively. The apparent half-life was 1.4 h and 1.9 h for lidocaine and mepivacaine, respectively, and the elimination of mepivacaine was thus slightly longer than of lidocaine.

## 4. Discussion

Local anesthetics in soft-tissue fillers typically represent 0.3% *w/w* of the composition of a commercial filler and are of paramount importance for pain management during and shortly after the injection procedure, and thus have an impact on patient satisfaction. Nowadays, almost all the manufacturers commercialize their products with an anesthetic, lidocaine being the most frequently used. Moreover, the addition of lidocaine in fillers does not impact the gel’s safety and effectiveness profiles [1,2], and the HA gel network does not influence lidocaine’s pharmacokinetics [24]. Here, a series of comparative investigations has been conducted on PNT-1 and PNT-4 fillers formulated with one of two different anesthetics, lidocaine and mepivacaine. The rheology of the fillers, which is widely used to characterize products and sort them into specific indications, was assessed in order to evaluate any influence of each anesthetic on the mechanical performance of the finished gels. In vitro and in vivo biological evaluations were also conducted to observe any impact of the anesthetics on the degradation profiles of the gels.

PNT-1 and PNT-4 presented distinct rheological properties specifically adapted to their respective intended clinical uses. In neither case did lidocaine nor mepivacaine significantly change the properties of the finished gels: all the rheological parameters of each gel remained well within their specifications and close to each other. The slight variations were more likely inherent to the inter-batch variability of the gel manufacturing. Before releasing a gel on the market, a series of different standardized testing must be conducted to monitor three key attributes: stability, safety, and performance. In terms of stability, the real-time shelf-life stability was assessed according to ICH guidelines. Soft-tissue fillers are designed to be stable over 2 to 3 years. In other words, their specifications, including mechanical properties, must remain within the specified values at least over the shelf-life specified on the product’s packaging to guarantee optimal and consistent clinical outcomes and safety. Despite the fact that it is well established that HA is highly sensitive to chemical, biological, and physical stimulations (pH, temperature, oxidative stresses, UV light, and enzymes), the properties can be stabilized through chemical crosslinking [25].

Via the comparative 36-month stability study, of which the results are presented here, mepivacaine-containing gels exhibited no difference in behavior when compared to lidocaine-containing gels, which confirms the compatibility of mepivacaine with long-term storage at room temperature. The limited variation of the gel’s rheological properties over the years remained well within the product specifications to ensure optimal outcomes whatever the date of use within the use-by date and is thoroughly monitored by manufacturers.

In terms of safety, the products were of course tested via a library of biocompatibility preclinical tests with respect to ISO 10993 standards, and afterwards by clinical trials. Even though possible but rare side effects can occur, a plethora of literature evidence demonstrates that HA fillers have a favorable benefit/risk profile [23,26,27]. The incorporation of lidocaine was also already proven to not adversely affect the safety profile of soft-tissue fillers [1,12]. Here, we demonstrated that there was no influence of the anesthetics, lidocaine or mepivacaine, in the preclinical susceptibility of the gels to be degraded by the hyaluronidase, traditionally used by practitioners as the standard antidote to HA fillers in case of adverse events, and which is the endogenous and predominant degrading enzyme of hyaluronic acid [25,28]. Mepivacaine thus acts as a reliable candidate to be incorporated in soft-tissue fillers for pain relief when considering these properties, as already demonstrated in dentistry [16], along with the fact it confers a reduced risk of systemic toxic reactions due to its lower lipid solubility (1.322 at pH = 7.4) compared to lidocaine (1.633 at pH = 7.4) [29]. The larger risk related to the use of anesthetics in fillers actually concerns patients suffering from allergic reactions in contact to amid-type anesthetics, which remains extremely rare [30].

In terms of performance, there was no difference in the impact of mepivacaine on the rheological properties of the gel compared to the lidocaine version. There was only a nonsignificant trend in the pharmacokinetics of these anesthetics favoring mepivacaine. It was notably observed that lidocaine reaches its maximum released concentration at a slightly earlier time compared to mepivacaine, with no statistical differences. Furthermore, mepivacaine presents a slightly longer half-life, meaning that the drug has to some extent a longer duration of action, in line with the literature [31]. This might be due to the consequence that mepivacaine exerts a less potent vasodilatory effect compared to lidocaine, thus resulting in less systemic absorption [5,16,17]. In addition, it is noteworthy that the values of mean plasma concentration are somewhat lower compared to the administered amount, in the order of 0.3 µg/mL for the maximum concentration observed. This is in line with the role of these local anesthetics, which should act around the injection site and not diffuse into the systemic circulation, before being naturally metabolized via several biotransformations [32,33].

These preclinical data were further confirmed by two randomized, double-blinded clinical trials proving the noninferiority of mepivacaine compared to lidocaine in terms of pain reduction. The clinical gel performance and safety were proven to be unaffected by the presence of lidocaine or mepivacaine [34]. Efforts were also being put forth to analyze the stability of mepivacaine itself in gels, as well as its potential degradation products via the development of analytical methods. These data will be the object of a separate publication.

## 5. Conclusions

Lidocaine and mepivacaine are two effective and helpful short-acting amide local anesthetics commonly used for topical and local anesthesia. They allow for intraoperative anesthesia and analgesia with a rapid onset and a convenient duration of action suitable for pain management during and in the period after the injection of fillers. The use of mepivacaine instead of lidocaine in the tested gels, adapted for intradermal and subcutaneous injection indications, did not impair their characteristics and properties in terms of preclinical safety, stability, and performance. Furthermore, no meaningful differences were detected when one anesthetic was used instead of the other, suggesting there were no issues in the use of one of these drugs in hyaluronic acid soft-tissue filler such as the RHA collection. Considering there is some evidence suggesting that mepivacaine has a lower vasodilatory activity than lidocaine, there may be clinical benefits making it an appropriate candidate to replace lidocaine in pain relief during hyaluronic acid filler injection.

## Figures and Tables

**Figure 1 pharmaceutics-14-01553-f001:**
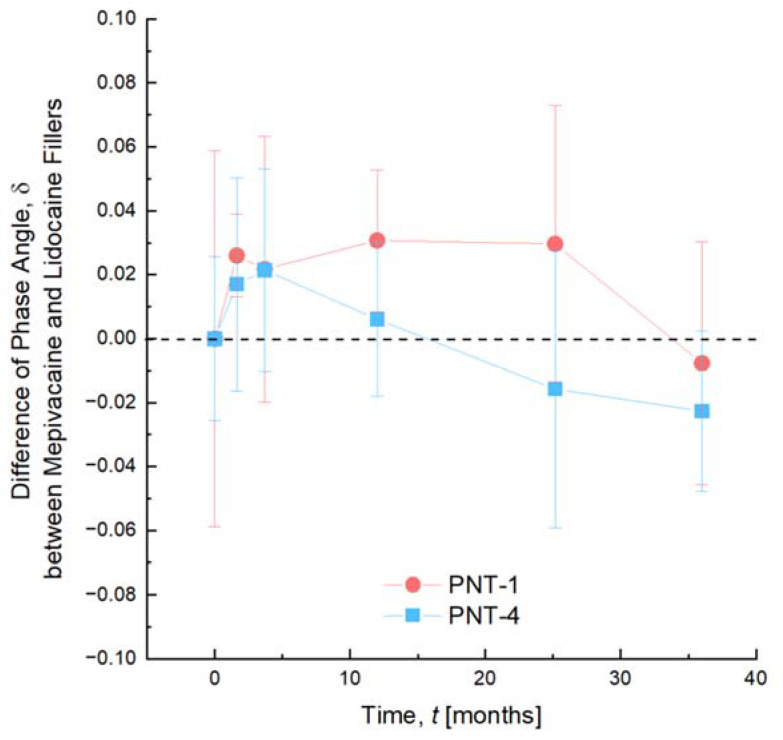
Shelf-life stability of PNT-1 and PNT-4 in the presence of lidocaine and mepivacaine over 3 years. The difference of phase angle between mepivacaine and lidocaine groups was monitored. At each timepoint, 3 syringes were evaluated for each group. Results are expressed as means ± SD. No significant variations were observed when comparing the products with either anesthetic, since the difference remained centered around 0. In each case, both groups maintained their rheological properties within their specifications over 3 years of storage, which makes these fillers deliver their optimal clinical outcomes within the use-by date.

**Figure 2 pharmaceutics-14-01553-f002:**
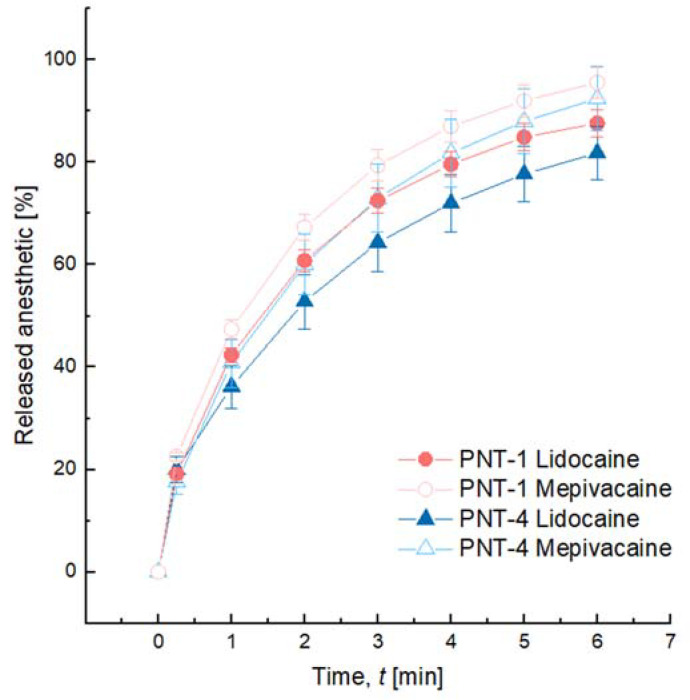
Release profiles of lidocaine and mepivacaine from PNT-1 and PNT-4. For each time point, and for each formulation, 3 gels were evaluated. Results are expressed as means ± SD. Complete release is almost reached after 6 h in the conditions of the test. Lidocaine and mepivacaine release profiles showed very similar trends and were considered equivalent, since no significance was observed between the different formulations at each time point. The significance was evaluated with a Fisher test F2.

**Figure 3 pharmaceutics-14-01553-f003:**
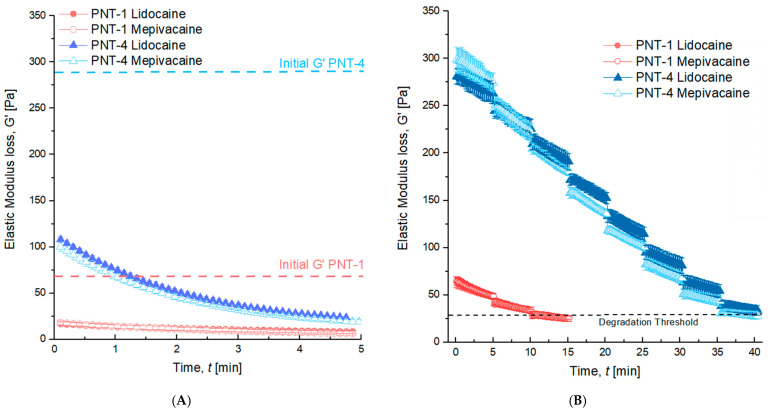
Enzymatic degradation test of PNT-1 and PNT-4 in presence of lidocaine or mepivacaine. (**A**) Fast degradation test with a high enzyme dose. The test highlighted the rapidness of degradation (5 min) of PNT-1 and PNT-4 with no influence of the anesthetic, which is mandatory for these kinds of medical devices in case of adverse events. (**B**) Persistence test with multiple low enzyme doses. This test showed similar behaviors of the gels either with lidocaine or mepivacaine, suggesting a similar clinical duration for each formulation either with lidocaine or mepivacaine. Each test was performed on 3 different HA gels for each formulation (*n* = 3). No significance was noticed between lidocaine and mepivacaine formulations.

**Figure 4 pharmaceutics-14-01553-f004:**
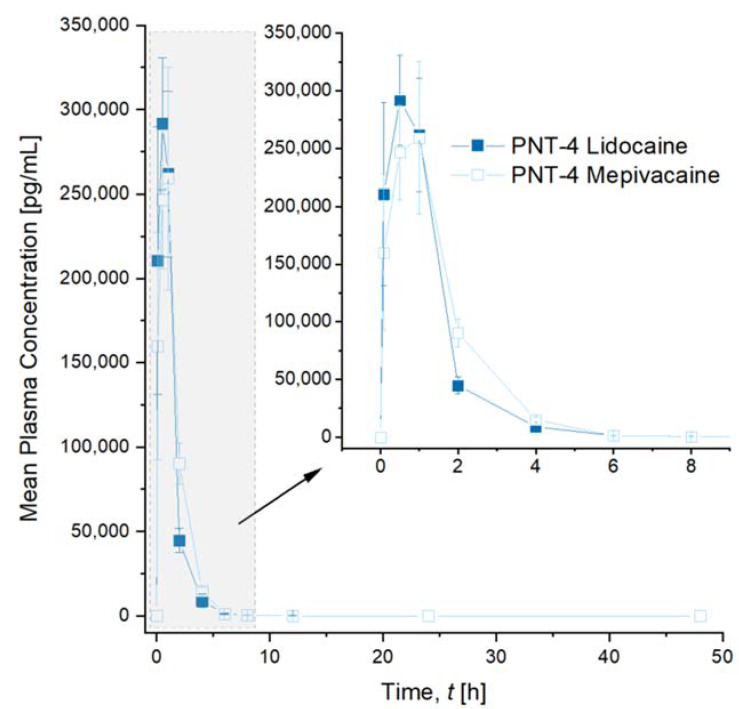
Pharmacokinetics curves of lidocaine and mepivacaine in rat plasma from PNT-4. PNT-4 gels were injected intradermally into rats. After a quick release of anesthetic in the plasma in the first hour, a rapid decrease of the plasma concentrations of both anesthetics was observed for 6 h. Lidocaine appeared in the plasma first and reached a higher maximum concentration compared to mepivacaine, while the mepivacaine plasma concentration was sustained for a longer duration compared to lidocaine. However, no statistical differences were observed between formulations at any timepoint. Three repeated measurements (*n* = 3) were carried out for each condition. Results are expressed as means ± SD.

**Table 1 pharmaceutics-14-01553-t001:** Intended use and composition of the investigated gels.

Product Abbreviation	Indications (May Differ Depending on the Country and Local Market Approvals; You Can Refer to the IFU for Further Details)	(HA) (mg/mL)	Degree of Modification (%)	Anesthetic Content (% *w*/*w*)
PNT-1	Superficial dynamic filler	15	2.0	0.3
PNT-4	Dynamic volumizer	23	4.0	0.3

**Table 2 pharmaceutics-14-01553-t002:** Mechanical properties of the investigated gels.

Product Name	Anesthetic	Strength (Elastic Modulus G′; LVER) (Pa; Pa)	Measured Phase Angle, δ (°)	Extrusion Force (*n*)	Stretch (10^−6^ s^−1^)
PNT-1	Lidocaine	69 ± 5; 76 ± 6	21.0 ± 0.9	13.9 ± 0.9	952 ± 206
	Mepivacaine	76 ± 2; 80 ± 1	21.3 ± 0.5	15.7 ± 0.9	1080 ± 29
PNT-4	Lidocaine	262 ± 11; 306 ± 9	7.3 ± 0.5	9.5 ± 0.2	49 ± 3
	Mepivacaine	259 ± 3; 300 ± 2	6.9 ± 0.0	9.6 ± 0.3	48 ± 9

**Table 3 pharmaceutics-14-01553-t003:** Pharmacokinetic parameters for lidocaine and mepivacaine after intradermal injection of PNT-4 with lidocaine and mepivacaine in rats.

Treatment	T1/2 (h)	TMAX (h)	CMAX (ng/mL)	AUC (ng. h/mL)
PNT-4 with lidocaine	1.4	0.50	291.7	430.2
PNT-4 with mepivacaine	1.9	1.00	259.3	475.4

## Data Availability

Data sharing is not applicable to this article.

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
