# Peer review of "Comparative Preclinical Study of Lidocaine and Mepivacaine in Resilient Hyaluronic Acid Fillers"

_pharmaceutics, 2022, doi:10.3390/pharmaceutics14081553_

Round 1

Reviewer 1 Report

This manuscript reports the rheological and pharmacokinetic properties of anesthetic included gels from hyaluronic acid (HA) for aesthetic dermal injection.  The authors want to find whether changing the anesthetic component from lidocaine to mepivacaine would cause any change to the HA-based gels or not.  The rationale behind this study is that the inclusion of mepivacaine into HA gels might offer better benefits than the more commonly used lidocaine because mepivacaine itself induces less widening of blood vessels than lidocaine.  The results showed no difference in rheology and concentration profile of anesthetic in rat plasma between lidocaine-included and mepivacaine-included HA gels.  The authors concluded that, “the use of either Lidocaine or Mepivacaine does not impact the properties of the gels” (Line 21-22), and “the use of mepivacaine instead of lidocaine in the tested gels, …, did not impair their characteristics and properties in terms of preclinical safety, stability, and performance” (Line 335-338).

The reviewer agrees that the results from this unprecedented study suggest the feasibility of substituting lidocaine with mepivacaine in HA gels.  However, the reviewer feels concerned about the following issues;

1.      there were no rheological results on control sample (i.e., HA gels without any anesthetic).  As a result, the conclusion that the use of either lidocaine or mepivacaine does not affect HA gels becomes overclaiming.

2.     there were several reports on the combinatorial effect of hyaluronidase and mepivacaine, which might complicate the results from the enzymatic degradation test in Figure 3.  The author should address whether hyaluronidase and mepivacaine have any physicochemical interaction or not.

3.      the extrusion force of PNT-1 is clearly higher than PNT-4 when other data suggests that PNT-1 is more liquid-like than the stiffer PNT-4.

Minor Issues

1.      The decimal separator on the x-axis in Figure 1 is a comma (,) instead of a dot (.).  However, a dot is used as the decimal separator in the rest of data in the paper.

2.      It remains unclear why these preclinical studies on rheology and drug release are necessary when studying the vasodilation ability of mepivacaine-included gels can really prove whether it is worth to substitute lidocaine with mepivacaine or not.

Author Response

First all of, all the authors want to kindly thank the editor for having the opportunity to revise our manuscript entitled “Comparative Preclinical Study of Lidocaine and Mepivacaine in Resilient Hyaluronic Acid Fillers”. We would additionally like to thank the reviewers for their useful and relevant comments.

To improve the overall quality of the manuscript, we have brought diverse revisions on the manuscript, as stated in the following answers.

Reviewer 1:

This manuscript reports the rheological and pharmacokinetic properties of anesthetic included gels from hyaluronic acid (HA) for aesthetic dermal injection.  The authors want to find whether changing the anesthetic component from lidocaine to mepivacaine would cause any change to the HA-based gels or not.  The rationale behind this study is that the inclusion of mepivacaine into HA gels might offer better benefits than the more commonly used lidocaine because mepivacaine itself induces less widening of blood vessels than lidocaine.  The results showed no difference in rheology and concentration profile of anesthetic in rat plasma between lidocaine-included and mepivacaine-included HA gels.  The authors concluded that, “the use of either Lidocaine or Mepivacaine does not impact the properties of the gels” (Line 21-22), and “the use of mepivacaine instead of lidocaine in the tested gels, …, did not impair their characteristics and properties in terms of preclinical safety, stability, and performance” (Line 335-338).

The reviewer agrees that the results from this unprecedented study suggest the feasibility of substituting lidocaine with mepivacaine in HA gels.  However, the reviewer feels concerned about the following issues;

  1. there were no rheological results on control sample (i.e., HA gels without any anesthetic).  As a result, the conclusion that the use of either lidocaine or mepivacaine does not affect HA gels becomes overclaiming.

We firstly would like to thank the reviewer for this relevant comment.

Through the conclusive sentence as mentioned by the reviewer, the authors wanted to convey the message that the mechanical properties of the gels incorporating mepivacaine were the same than the properties of the gels incorporating lidocaine.

Indeed, nowadays lidocaine-based fillers have become the norm in the current soft-tissue filler injection practice. To prove this, the commercial Teosyal RHA product line is only offered with a lidocaine version and no products are available without anesthetics. As a results, we were not able to provide data about RHA fillers PNT-1 and PNT-4 without lidocaine as requested by the reviewer since they are not manufactured.

We thus decided to focus our comparison with only lidocaine-based fillers which are the market references.

The conclusion was then effectively not properly formulated to match the message claimed.
So, in accordance with the reviewer’s point, we rephrased our conclusion as follows:

“Substituting Lidocaine with mepivacaine does not impact the properties of the gels, thus both can be equally incorporated as anesthetics in soft-tissue fillers.” Lines 23-24

  1. there were several reports on the combinatorial effect of hyaluronidase and mepivacaine, which might complicate the results from the enzymatic degradation test in Figure 3.  The author should address whether hyaluronidase and mepivacaine have any physicochemical interaction or not.

We perfectly understand the reviewer’s point.

The investigation of the present study was intended to highlight any negative impact of the use of Lidocaine or Mepivacaine on the gel manufacturing or storage, and consequently, gel performance. In practice during aesthetic surgery procedures, if the use of Hyaluronidase is requested by the practitioner in order to correct any adverse events related to the gel - such as occlusion of a blood vessel, it generally occurs several hours, days or even months after the gel implantation, timeline which is longer than anesthetics residence time (max 2-4 hours). So, Hyaluronidase is not even legitimately used in presence of lidocaine or mepivacaine.
In addition, what was demonstrated in this article factually demonstrates that independently of any interaction, hyaluronidase still degrades mepivacaine-incorporating gels, similarly to how it degrades lidocaine-incorporating gels.

Besides, Hyaluronidase is an enzyme which is mainly recommended to temporarily break down extracellular matrix to facilitate API diffusion/permeability in tissues such as Lidocaine and Mepivacaine as indicated in the Indications For Use. Thus, Hyaluronidase acts as a permeability enhancer, but, to the best of our knowledge, do not physicochemically interact with APIs themselves. For instance, Nathan et al. in “The role of hyaluronidase on lidocaine and bupivacaine pharmacokinetics after peribulbar blockade” (1996) showed that Hyaluronidase boosted the absorption kinetics of both lidocaine and bupivacaine (a mepivacaine analogue) in the same manner. Similarly, Remy et al. in “Efficacy and safety of hyaluronidase 75 IU as an adjuvant to mepivacaine for retrobulbar anesthesia in cataract surgery” (2008) demonstrated that the use of Hyaluronidase hastened the onset of total anesthesia.

  1. the extrusion force of PNT-1 is clearly higher than PNT-4 when other data suggests that PNT-1 is more liquid-like than the stiffer PNT-4.

The authors thank the reviewer for this comment and want to apologize since we forgot to precise the needle size used when measuring the extrusion force of each filler.

The needles used in this study were provided in each filler box.

Indeed, PNT-1 which is a soft filler is prescribed to be injected through a 30G needle, in contrary to PNT-4 which is a volumizer and should be injected though a 27G needle, thus explaining the indicated results. If we compare the extrusion forces of both fillers using the same 27G needle, the extrusion force of PNT-1 would definitely be lower than that of PNT-4.

A sentence was added accordingly into the text (l 92-93):” The injection force was measured using a force tester (MultiTest-dV, Mecmesin) using 30 G ½’ and 27 G ½’ needles for PNT-1 and PNT-4, respectively, at a constant speed of 12.5 mm/s.”

Minor Issues

  1. The decimal separator on the x-axis in Figure 1 is a comma (,) instead of a dot (.).  However, a dot is used as the decimal separator in the rest of data in the paper.

We acknowledge the reviewer for pointing out this typo which was corrected in Figure 1.

  1. It remains unclear why these preclinical studies on rheology and drug release are necessary when studying the vasodilation ability of mepivacaine-included gels can really prove whether it is worth to substitute lidocaine with mepivacaine or not.

We understand the reviewer’s concern.

The presented preclinical studies, conducted in this article, are part of the preliminary regulatory study a manufacturer needs to present to Authorities for market acceptance such as in the case in the use of a new anesthetics. A clinical trial was also conducted and published (Kaufman-Janette et al.,Patient Comfort, Safety, and Effectiveness of Resilient Hyaluronic Acid fillers formulated with different Local Anesthetics” Accepted in Dermatologic Surgery (2022)) about this topic.

These investigations are thus important to the eye of Health Care Practitioners and Regulatory Authorities.

In this manuscript, we indeed also stated the potential clinical benefit of mepivacaine versus lidocaine from the literature. Nonetheless, the point of this article was to only study preclinical properties of these fillers.

Efforts are also currently being put forth to analyze the ex vivo and in vivo vasodilatory activity of mepivacaine and lidocaine. These data being of meaningful importance for practitioners, they will be the object of a separate publication.

Reviewer 2 Report

The study is intended to compare two drug substances (Lidocaine and Mepivacaine) for their use as anesthetics in aesthetic procedures for correction of skin tissue defects and volume loss. The study is well structured and methodology is well described. The data are clearly presented and the results are supporting the conclusions. There is a major difference between the mechanical properties of gels - the Stretch around 1000  (.10-6 s -1) for PNT-1 and around 50  (.10-6 s -1) for PNT-4 (Table 2), a difference which is not explained or discussed. What is not evident for me is that Table 1 presents the two gels (PNT-1 and PNT-4) having different functions: PNT-1 is  a superficial dynamic filler and PNT-4 is a dynamic volumizer. In all experiments they were tested separately and not together. My question is if in practice the two will be used together or not ? If YES, they should also be mixed and their performance (in vitro and in vivo) analyzed. There are few minor corrections to be done: line 125 in vitro to be changed in italics; Fig 2: Time units are hours not minutes.

Author Response

First all of, all the authors want to kindly thank the editor for having the opportunity to revise our manuscript entitled “Comparative Preclinical Study of Lidocaine and Mepivacaine in Resilient Hyaluronic Acid Fillers”. We would additionally like to thank the reviewers for their useful and relevant comments.

Reviewer 2:

the study is intended to compare two drug substances (Lidocaine and Mepivacaine) for their use as anesthetics in aesthetic procedures for correction of skin tissue defects and volume loss. The study is well structured and methodology is well described. The data are clearly presented and the results are supporting the conclusions. There is a major difference between the mechanical properties of gels - the Stretch around 1000  (.10-6 s -1) for PNT-1 and around 50  (.10-6 s -1) for PNT-4 (Table 2), a difference which is not explained or discussed. What is not evident for me is that Table 1 presents the two gels (PNT-1 and PNT-4) having different functions: PNT-1 is  a superficial dynamic filler and PNT-4 is a dynamic volumizer. In all experiments they were tested separately and not together. My question is if in practice the two will be used together or not ? If YES, they should also be mixed and their performance (in vitro and in vivo) analyzed. There are few minor corrections to be done: line 125 in vitro to be changed in italics; Fig 2: Time units are hours not minutes.

Authors: In the first place, we would like to thank the reviewer for its comment.

The focus of the paper was the comparison between lidocaine-incorporating fillers and mepivacaine-incorporating fillers.

Besides, as now presented in paragraph §2.1, both products are prescribed for two well separated clinical indications (l77-78):” HA-based fillers PNT-1, intended to correct superficial fine lines, and PNT-4, used subcutaneously or in deep fat compartments as a volumizer, were chosen among the Teosyal RHA® collection manufactured by Teoxane SA, Switzerland.”

As a result, both products have very distinct mechanical properties adapted for very different clinical indications. In consequence, in medical practice, both fillers will never be used together for the same treatment, and it would also be an off-label use to do such mix up. That is why they were studied separately, as fillers with opposed behaviors.

However, to facilitate the readers’ comprehension, following the reviewer’s remarks, we included sentences in the manuscript to clarify the mechanical properties of each type of fillers and their differences.

The revisions are presented as follows:

3.1 – Lines 162-166: “[…] the more the gel may accommodate more naturally and rapidly to facial expressiveness. The Strength score as well as the G’ were higher for PNT-4 than PNT-1, meaning that PNT-4 presented a higher elastic behavior, suitable for deeper indications to lift tissues. PNT-1 on the opposite presented a higher Stretch score compared to PNT-4 which confirmed that PNT-1 is well adapted to the superficial dynamic areas of the skin. Upon comparison of lidocaine and mepivacaine, […]”

Round 2

Reviewer 1 Report

The reviewer would like to thank the authors for responding with clear explanations to the reviewer's comments on scientific issues. 

After re-reading the second-version manuscript, the reviewer thinks the author should include the author's responds to the reviewer's comment into the text. 

A) the reason for preclinical studies to the last introduction paragraph (line 66-71)

B) the validity of using hyaluronidase in this studies to the discussion section.

It would help readers understand more about the importance and research design of this study.  Other than that, after a thorough check on grammar and language flow, this manuscript can be ready for publishing. 

Author Response

Second round of revisions of Comparative Preclinical Study of Lidocaine and Mepivacaine in Resilient Hyaluronic Acid Fillers:

The authors: We would like to kindly thank again the editor and reviewers for their useful comments and for the opportunity to revise our manuscript entitled “Comparative Preclinical Study of Lidocaine and Mepivacaine in Resilient Hyaluronic Acid Fillers”.

On behalf of the last suggestions, we have brought some revisions on the manuscript, as stated in the following answers.

Reviewer 1:

The reviewer would like to thank the authors for responding with clear explanations to the reviewer's comments on scientific issues. 

After re-reading the second-version manuscript, the reviewer thinks the author should include the author's responds to the reviewer's comment into the text. 

  1. the reason for preclinical studies to the last introduction paragraph (line 66-71)

The authors: We agree with the reviewer’s comment to emphasize the reason of such a study.

In consequence, in lines 66-73, we added (highlighted in blue):

 “In this study, mepivacaine was investigated as a potential new local anesthetic agent to be used in HA fillers to replace lidocaine. The choice of mepivacaine is notably encouraged by its lower vasodilatory activity compared to lidocaine, mepivacaine tending to either preserve or decrease peripheral blood flow, keeping a lower systemic concentration of mepivacaine over time, which would turn out to be an additional safety aspect [16-18]. The influence of this anesthetic agent was then compared to the gold standard lidocaine in terms of filler mechanical properties, stability, degradability, release profiles, and pharmacokinetics. Such data are especially useful for a better evaluation of the safety and the performance of mepivacaine-loaded fillers compared to their lidocaine counterparts and, as such, are relevant for the registration of the device.”

The reviewer:

B) the validity of using hyaluronidase in this studies to the discussion section.

The authors:

We acknowledge the reviewer’s comment.

The choice of using hyaluronidase to degrade fillers is an interesting point mentioned by the reviewer since hyaluronic acid, which is the main ingredient of fillers and as a polysaccharide, is susceptible to degrade in contact to various types of enzymes. Nevertheless, this is commonly reported that endogenous hyaluronidases are the main family of enzyme involved in hyaluronic acid degradation as presented for example by Stern et al. in “The many ways to cleave Hyaluronan” (2007), Biotechnology Advances, and the mechanism is also well described by Stern in “Hyaluronan catabolism: a new metabolic pathway” (2004) European Journal of Cell Biology.

Moreover, as mentioned line 271, commercial forms of hyaluronidase are in practice used by Health Care Practitioners on a regular basis to treat adverse events.

Nonetheless, to precise the validity of using hyaluronidase in this study, we thus added the following sentence in the manuscript (highlighted in blue):
Lines 270 – 273: “Here, we demonstrated that there was no influence of the anesthetics, lidocaine or mepivacaine, in the preclinical susceptibility of the gels to be degraded by the hyaluronidase, traditionally used by practitioners as the standard antidote to HA fillers in case of adverse events and which is the endogenous and predominant degrading enzyme of hyaluronic acid [25-28].”